# Conformational plasticity of the HIV-1 gp41 immunodominant region is recognized by multiple non-neutralizing antibodies

Jonathan D. Cook [1,2], Adree Khondker[1] & Jeffrey E. Lee [1✉]

The early humoral immune response to acute HIV-1 infection is largely non-neutralizing. The principal target of these antibodies is the primary immunodominant region (PID) on the gp41 fusion protein. The PID is a highly conserved 15-residue region displayed on the surface of HIV-1 virions. In this study, we analyzed the humoral determinants of HIV-1 gp41 PID binding using biophysical, structural, and computational methods. In complex with a patient-derived near-germline antibody fragment, the PID motif adopts an elongated random coil, whereas the PID bound to affinity-matured Fab adopts a strand-turn-helix conformation. Molecular dynamics simulations showed that the PID is structurally plastic suggesting that the PID can form an ensemble of structural states recognized by various non-neutralizing antibodies, facilitating HIV-1 immunodominance observed in acute and chronic HIV-1 infections. An improved understanding of how the HIV-1 gp41 PID misdirects the early humoral response should guide the development of an effective HIV-1 vaccine.

[1] Department of Laboratory Medicine and Pathobiology, Temerty Faculty of Medicine, University of Toronto, Toronto, ON M5S 1A8, Canada. [2] Present address: Department of Medicine, Temerty Faculty of Medicine, University of Toronto, Toronto, ON M5S 1A8, Canada. ✉email: jeff.lee@utoronto.ca

Failure to mount an effective humoral immune response against the HIV-1 envelope glycoprotein (gp160) has been observed in the majority of vaccine trials to date[1] and is a hallmark of the natural history of HIV-1 infection. HIV-1 gp160 exists in the viral membrane as a trimeric complex of non-covalently attached gp120-gp41 heterodimers. Owing to the non-covalent and unstable nature of gp160, the gp120 attachment subunit is shed from the native trimers inducing a post-fusion conformation of gp41 on the surface of the mature virus[2,3]. It has been observed that only 7–14 functional trimeric gp160 spikes exist on the virion surface[4,5] and the majority of gp160 have lost gp120 to form fusion-incompetent gp41 stumps[6]. These post-fusion gp41 stumps produce a hairpin structure with likely both its fusion peptide and transmembrane anchor inserted into the viral membrane, thus exposing the chain reversal region. Likely owing to the abundance of gp41 accessible to the host humoral immune system, anti-gp41 antibodies are detectable at the earliest stages of HIV-1 infection and persist in those afflicted with chronic infection[7–10]. The vast majority of these antibodies are considered non-neutralizing and do not block HIV-1 entry or fusion. These antibodies can elicit some antibody-dependent cellular cytotoxicity, which was determined to be responsible for the modest protection observed in the RV144 vaccine trial[11]. However, antibody-dependent cellular cytotoxicity alone is insufficient for robust prophylaxis against a productive infection[12–14].

Multiple anti-gp41 monoclonal antibodies that specifically target a primary immunodominant region (PID) located within the gp41 ectodomain have been isolated from patient sera. The PID is an amphipathic 15-amino-acid region that separates the two heptad-repeat domains found in the post-fusion conformation of gp41. The PID is flanked by tryptophan residues 596 and 610 and has an internal disulfide bond between C598 and C604. The sequence of the PID is highly conserved across HIV subtypes, and as the moniker suggests, it is particularly immunodominant. In one human study, ~70% of all antibodies generated in an acute HIV-1 infection were toward an HIV-1 gp41 PID-containing peptide[10]. Of note, the PID is not fully accessible in the gp160 pre-fusion conformation[15], suggesting that anti-HIV-1 PID antibodies may predominantly recognize the epitope in the context of virions with gp41 spikes. This hypothesis is supported by multiple studies that have shown non-neutralizing anti-PID antibodies can capture freely circulating HIV-1 virions[16,17]. Partitioning of viral particles by epitope recognition revealed that HIV-1 virions sort into subpopulations of infectious and non-infectious particles. There are two subsets of infectious virions: those that are almost exclusively bound by neutralizing Env antibodies, and those that are bound by both neutralizing and non-neutralizing antibodies such as those that target the PID. The non-infectious population is readily bound by anti-PID antibodies but is not bound by neutralizing Env antibodies[6]. How the PID motif misdirects the adaptive immune system to elicit non-neutralizing antibodies is poorly understood.

In pursuit of a structural understanding of the humoral determinants of HIV-1 gp41 PID immunodominance, we employed an integrative structural, biochemical, and computational approach to study how HIV-1 interacts with the immune system. We determined the high-resolution crystallographic structures and performed molecular dynamics (MD) simulations on two patient-derived Fabs unbound and in complex with the gp41 PID epitope. Our high-resolution crystallographic models and MD simulations revealed that the PID is conformationally plastic with multiple conformational states. Our results provide insight into the mechanism of HIV-1 gp41 immunodominance, contribute to our understanding of the diversity of mechanisms that HIV-1 uses to escape immune surveillance, and provide guidance for future HIV-1 vaccination efforts.

## Results

**Characterization of interactions between Cluster I antibodies and gp41.** Cluster I anti-gp41 antibodies, including patient-derived 3D6, 7B2, and F240, bind to the HIV-1 PID region (HXB2 isolate residues: 596–610). Genetic analysis of Cluster I antibodies suggests that they are representative members of a clonotypic response from the adaptive immune system[17,18]. 3D6 has a near-germline configuration, and F240 and 7B2 represent anti-PID antibodies that have undergone affinity maturation. Using purified recombinant antibody fragments (Fab) of 3D6 and F240, we performed interaction studies between the Fabs and an overlapping peptide set consisting of 15-mer peptides that span the PID region within the HIV-1 gp41 ectodomain. As expected, the PID clonotypic epitope was mapped by enzyme-linked immunoassay (ELISA) to a single peptide that corresponds to residues 594–608 of HIV-1 gp160 (Fig. 1 and Supplementary Data 1).

**Structures of patient-derived Cluster I antibody fragments complexed to PID.** To investigate the determinants of the host anti-PID response via structural biology, crystal structures of the 3D6 and F240 anti-gp41 PID antibody fragments unbound and bound to a 15-mer HIV-1 gp41 PID peptide were determined (Table 1). The overall structures of Fab 3D6 and F240 bound to the HIV-1 gp41 PID motif showed well-defined electron densities for the residues of the PID (Supplementary Fig. 1).

The PID peptide bound to Fab 3D6 adopts an elongated random coil conformation skewed toward the antibody heavy chain (Fig. 2). The binding of PID to Fab 3D6 is achieved primarily via interactions between residues 601–610 of the PID peptide and the loops of the three complementarity-determining regions (CDRs) of the heavy chain. The interactions of PID with the heavy chain occludes 550 $\text{Å}^2$ surface area, whereas interactions to the light chain only buries 170 $\text{Å}^2$ surface area through minor interactions with CDR-L1 and CDR-L3 loops. Specifically, CDR-H2 and CDR-H3 form the bulk of the van der Waals and hydrogen bonds interactions to PID. The interface culminates at the CDR-H2 loop, which contributes several hydrogen bonds to the coordination of the PID peptide via CDR-H2 residues W52A, D53, and S56. The CDR-H3 β-finger via Y100D makes two main-chain hydrogen bonds with the extended PID peptide, creating an extension of the β-finger into a pseudo-antiparallel three-stranded β-sheet.

In our unbound Fab 3D6 structure, CDR-H3 is mostly disordered, suggesting some intrinsic flexibility, however, primary sequence analysis using DEPICTER[19] does not identify a region of intrinsic disorder. Of note, this loop was ordered in the previously reported unbound structure of Fab 3D6[20], allowing for additional comparative analysis. From the 3D6 light chain, only residue N92 in the CDR-L3 loop forms a main-chain hydrogen bond to PID L602. Additional van der Waals interactions are formed between W32, S93, and Y94 to PID residues K601, L602, and I603, respectively.

The HIV-1 gp41 PID peptide in the context of Fab F240 adopts a β-strand-loop-helix motif (Fig. 3). The PID β-strand-loop-helix conformation is stabilized by intramolecular van der Waals interactions between the indole ring of W596 and the hydroxyl group of S604. In the full-length PID, an analogous interaction would be found between the indole and the sulfur of C604. In addition, the side chain of W596 packs into a hydrophobic pocket generated by A608 and V609, shielding these hydrophobic residues from the bulk solvent. The PID region is hypothesized to form a disulfide bond between C598 and C604[16]. In the case of the F240-bound PID peptide, the cysteine residues were mutated to serines; a 2.6-Å hydrogen bond between C598S and C604S

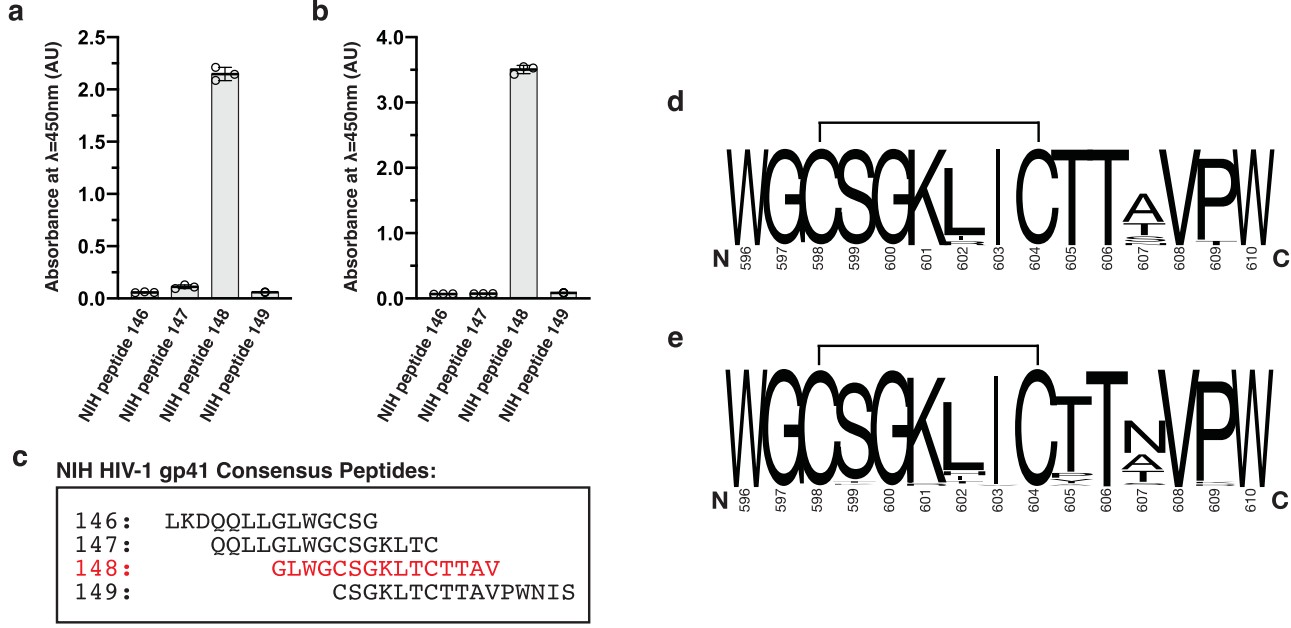

**Fig. 1 Epitope mapping of Fab F240 and Fab 3D6 by ELISA. a** Fab 3D6 and **b** Fab F240-binding profiles obtained by ELISA. The absorbances were measured as technical triplicates and individual data points shown in the bars ±standard deviation (SD). **c** Sequences of NIH HIV-1 gp41 peptides used in the ELISA study. **d** Logoplot of the consensus sequence of the PID motif from HIV-1 M group subtype B that is prevalent in North America and Europe. **e** Logoplot of the consensus sequences of the PID motif from all HIV-1 subtypes characterized[85,86]. C598 and C604 have been previously demonstrated to form a disulfide bond, depicted here by a solid line.

recapitulates the disulfide bond that would naturally exist in the PID region. These intramolecular interactions stabilize a compact structure of the HIV-1 gp41 PID epitope that interacts at the heavy and light chain interface. The interactions of PID with Fab F240 can be partitioned in half with the N-terminal PID β-strand (residues 596–601) interacting primarily with the heavy chain, and the C-terminal PID α-helix (residues 602–610) forming interactions with the light chain. Overall, the PID buries 397 Å$^2$ and 246 Å$^2$ of surface area on the heavy and light chains, respectively. All six CDRs (CDR-H1, CDR-H2, CDR-H3, CDR-L1, CDR-L2, and CDR-L3) contact the PID. Specifically, the PID β-strand interacts with the heavy chain CDR-H1 and CDR-H3 through the formation of an intermolecular β-sheet with a β-finger structural motif located in the CDR-H3. The majority of interactions are hydrogen bonds between main-chain carbonyl oxygen or amide nitrogen atoms of both the PID peptide and Fab F240, with the exception of side-chain hydrogen bonds from K52A and D96 to PID. The PID α-helix interacts with the light chain CDR-L1 and CDR-L2 through van der Waals forces and six hydrogen bonds.

**Interactions of Fab 3D6 with PID are near-germline.** Many anti-HIV-1 broadly neutralizing antibodies, such as the VRC01 class, contain numerous somatic mutations[21,22], with mutation frequencies of ~30% in the heavy chain and 20% in the light chain variable region genes[23]. In humans, affinity-matured antibodies are expected to contain ~10–20 mutations per chain[24], whereas broadly neutralizing antibodies against HIV-1 harbor four to five times that number and are often distributed throughout both CDR and framework regions[22]. Antibody F240 shows a considerable divergence from the germline sequence with 18.7% (54 nucleotides) and 15.1% (45 nucleotides) of heavy and light chain coding sequence mutated, respectively (Supplementary Fig. 2)[18]. In contrast, analysis of the Fab 3D6 cDNA sequence using the IMGT/V-QUEST tool[25] indicates that there have been few mutations in variable regions of the heavy and light chains of Fab

3D6, with only 2.1% (six nucleotides) and 1.1% (three nucleotides) of the heavy and light chain nucleotides mutated, respectively (Supplementary Fig. 2). These nine mutations result in five changes at the amino acid level, and of these mutations, only one contributes to the Fab 3D6 paratope (CDR-H2 D53). A single transition mutation in the inferred-germline configuration generates an asparagine to aspartic acid. However, hydrogen bonding between the Oδ of CDR-H2 residue D53 and the side chain of PID residue T605 would likely be conserved in the germline asparagine configuration. In the case of the anti-PID antibody response, the Fab 3D6 structures presented here support the notion that these non-neutralizing antibodies can rapidly arise from germline B-cell receptor (BCR) configurations and bind to gp41 spikes using preformed paratopes.

**3D6 and F240 display disparate mechanisms of interactions to the PID.** Comparison of the PID-bound and unbound structures of 3D6 and F240 revealed that the two Fab fragments have different mechanisms of binding (Fig. 4). For F240, no appreciable conformational differences were detected in any of the CDR loops in the heavy or light chains between bound and unbound conformations. For 3D6, the CDR-H3 undergoes considerable conformational change suggestive of an induced-fit mechanism of binding to the PID. Residues S100A, G100B, S100C, and Y100D from the CDR-H3 loop are disordered in the unbound state. Upon binding to the PID, residues 100B to 100 F extend to form a β-hairpin in the bound structure. The CDR-H3 loop moves 2.5 Å toward the CDR-H2 loop wedging the PID into a pocket generated by the FR2 C' β-strand. This movement breaks the π-π stacking interaction between the aromatic rings of $V_L$ residue W32 and $V_H$ residue F100E to accommodate PID residue K601. Flexibility in CDR-H3 loops is commonly seen in germline and near-germline antibodies, with progressive rigidification observed during affinity maturation[26–29]. This observed rigidity in matured CDR-H3 loops generally stabilizes the antigen-binding site in a conformation ideal for antigen interactions and often leads to

**Table 1 Data collection and refinement statistics for Fab 3D6 and Fab F240 crystal structures.**

|  | Apo-Fab 3D6 | Fab 3D6:PID complex | Apo-Fab F240 | Fab F240:PID complex |
|---|---|---|---|---|
| Wavelength (Å) | 0.9801 | 0.9795 | 0.9795 | 0.9795 |
| Resolution range (Å)[a] | 48.00–2.40(2.49–2.40) | 42.49–2.00 (2.03–2.00) | 44.71–1.70(1.76–1.70) | 45.70–1.70(1.76–1.70) |
| Space group | P4₃2₁2 | P32 | P2₁2₁2₁ | P2₁2₁2 |
| Unit cell |  |  |  |  |
| a, b, c (Å) | 84.4, 84.4, 161.0 | 77.8, 77.8, 218.9 | 47.0, 74.4, 145.0 | 46.1, 168.8, 78.4 |
| α, β, γ (°) | 90, 90, 90 | 90, 90, 120 | 90, 90, 90 | 90. 90, 90 |
| Total reflections[a] | 256,579 (27,661) | 198,503 (9,776) | 109,445 (8374) | 132,489 (10557) |
| Unique reflections[a] | 23,585 (2,430) | 98,127 (4,871) | 54,928 (4303) | 66,230 (5299) |
| Multiplicity[a] | 10.9 (11.4) | 2.0 (2.0) | 2.0 (1.9) | 2.0 (2.0) |
| Completeness (%)[a] | 100.0 (100.0) | 97.9 (98.6) | 96.6 (76.7) | 97.1 (78.4) |
| Mean I/sigma(I)[a] | 14.3 (1.9) | 8.3 (1.5) | 11.6 (1.0) | 16.7 (0.9) |
| Wilson B-factor | 47.0 | 23.8 | 23.8 | 24.6 |
| R-merge (%)[a,b] | 12.2 (143.6) | 7.7 (66.8) | 2.3 (5.6) | 2.7 (83.5) |
| CC1/2[a] | 0.999 (0.775) | 0.995 (0.551) | 1.0 (0.53) | 1.0 (0.30) |
| R-work (%)[a,c] | 19.5 (29.4) | 16.6 (25.4) | 17.9 (34.5) | 17.2 (36.8) |
| R-free (%)[a] | 23.9 (33.4)[d] | 20.3 (29.0)[e] | 22.2 (38.8)[f] | 20.4 (37.3)[g] |
| Number of non-hydrogen atoms | 3382 | 11,498 | 3763 | 3941 |
| Macromolecules | 3293 | 10,285 | 3410 | 3493 |
| Ligands | 10 | - | 12 | 19 |
| Solvent | 79 | 1,213 | 341 | 429 |
| RMSD (bonds; Å) | 0.010 | 0.011 | 0.010 | 0.010 |
| RMSD (angles; °) | 1.4 | 1.1 | 1.1 | 1.1 |
| Ramachandran favored (%) | 96.8 | 97.9 | 99.0 | 98.2 |
| Ramachandran allowed (%) | 3.0 | 2.1 | 1.1 | 1.8 |
| Ramachandran outliers (%) | 0.2 | 0 | 0 | 0 |
| Clashscore | 2.9 | 2.4 | 0.74 | 0.6 |
| Average B-factor | 32.6 | 29.5 | 34.4 | 31.6 |
| Macromolecules | 32.2 | 28.9 | 33.8 | 30.3 |
| Ligands | 93.0 | - | 63.6 | 32.3 |
| Solvent | 45.8 | 34.7 | 39.8 | 42.0 |
| Number of TLS groups | 11 | 45 | 18 | 12 |
| PDB accession number | 7N07 | 7N08 | 7N04 | 7N05 |

[a]Statistics for the highest-resolution shell are shown in parentheses.

[b]$R_{merge} = \Sigma\Sigma_j|I_j-<I>|/\Sigma|<I>|$, where $I_j$ and $<I>$ represent the diffraction intensity values of the individual measurements and the corresponding mean values, respectively. The summation is over all unique measurement.

[c]$R_{work}$: $= \Sigma||F_{obs}|-|F_{calc}||/\Sigma|F_{obs}|$, where $F_{calc}$ and $F_{obs}$ are the calculated and observed structure factor amplitudes, respectively.

$R_{free}$ statistic is the same as $R_{work}$ except calculated on [d]8.5%, [e]2.1%, [f]3.6%, and [g]3.0% of the total reflections chosen randomly and omitted from the refinement.

higher affinity interactions[29–31]. For germline CDR-H3 loops, there is in silico evidence that this flexibility can create an ensemble of conformations that accommodate diverse antigens at the cost of affinity[27,28,32,33]. Thus, it appears that F240 has a rigid antigen-binding site that recognizes the gp41 PID through a preformed lock-and-key mode of binding, whereas the near-germline antibody 3D6 binds PID through induced-fit. This contrasts with HIV-1 broadly neutralizing antibodies of the VRC01 class that does not exhibit this classical affinity maturation pathway[34].

There is another major difference in the PID binding modes between F240 and 3D6. In most anti-peptide antibody structures, the peptide antigen binds in a groove at the interface of the heavy and light chains. This mode of peptide binding is observed in Fab F240, where the PID peptide form interactions to both antibody chains. In contrast, 3D6 does not bind the PID at the interface of the heavy and light chains. Instead, the majority of interactions are formed solely between the heavy chain and the PID. Specifically, the PID peptide binds in a groove formed by CDR-H3 on one side and CDR-H1 and CDR-H2 on the other side (Fig. 2), reminiscent of known single-chain nanobody interactions[35].

**HIV-1 gp41 PID peptide displays multiple conformations.** The PID peptides bound to Fabs 3D6 and F240 have different conformations (Fig. 5). The PID peptide bound to Fab F240 forms a β-strand-loop-helix conformation. Interestingly, this conformation is similar to that observed in the complex with another patient-derived anti-gp41 PID antibody 7B2[16]. In both F240 and 7B2, the PID region forms a β-strand-loop-helix structure that interacts with the β-finger structural motif in the CDR-H3 loop of the Fab. In contrast, the extended PID conformation observed in the complex with Fab 3D6 is distinct from its conformation observed in the complexes with Fab F240 or Fab 7B2. The conformations of PID regions bound to F240, 7B2, and 3D6 are also different from the pre-fusion conformation in HIV-1 gp160[36]. The crystal structure of the pre-fusion HIV-1 SOSIP gp140 revealed that PID residues 602–606 form an intermolecular β-sheet with the gp120 subunit[37] (Supplementary Fig. 3). The PID motif is mostly buried in the core of the pre-fusion gp140 protein, with only residues 608–611 partially exposed, thus explaining why Fabs 3D6, F240, and 7B2 either do not or poorly recognize the pre-fusion trimeric Env conformation. The unliganded PID motif characterized in solution by NMR spectroscopy adopts a type I reverse turn flanked by a disulfide bond between C598 and C604[38], consistent with the conformations of PID bound to both Fab F240 and 7B2. In addition, when interrogating the biophysical characteristics of a reduced form of the PID peptide, the type I reverse turn was abolished. 3D6 readily binds a large subset of wild-type HIV-1 virions[6]. Taken together, these findings are consistent with our structure of 3D6-bound PID. The observed PID conformations in the Fab F240, 7B2, and 3D6-bound

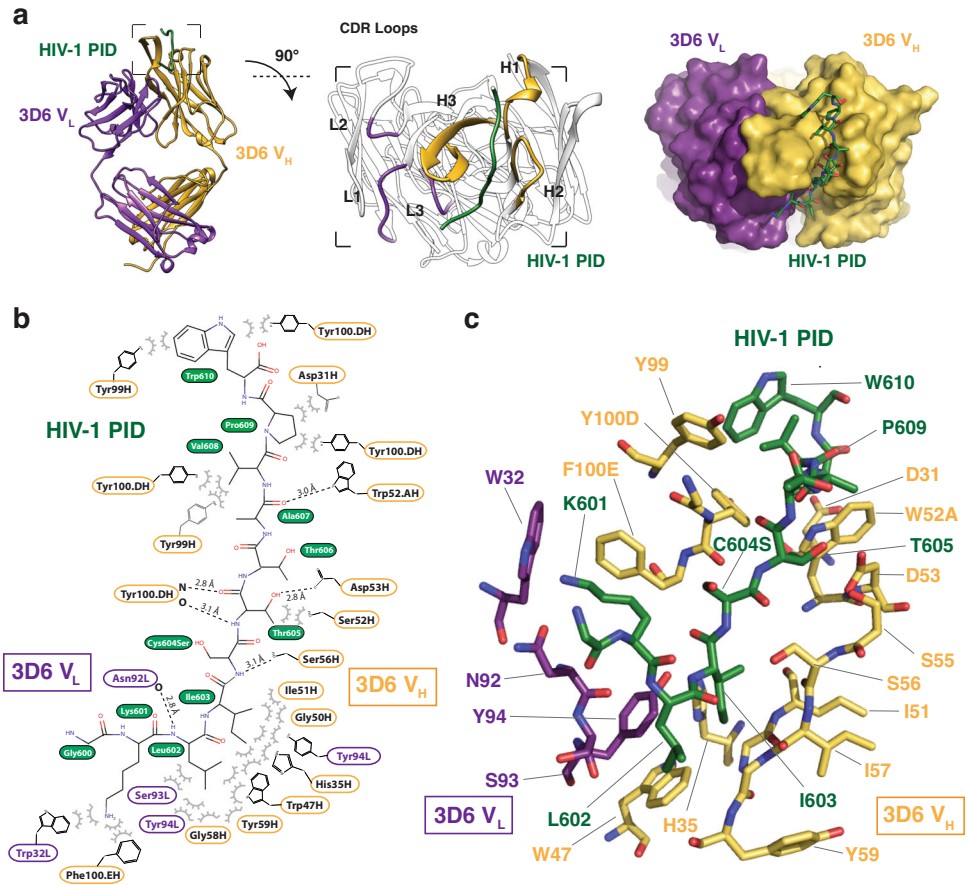

**Fig. 2 Overall structure of Fab 3D6 complexed to HIV-1 gp41 PID. a** The crystal structure of the Fab 3D6-PID complex shows the PID region in an extended conformation coordinated mainly by the $V_H$ subunit. Fab 3D6 CDR-H3 forms a pseudo-β-sheet with the HIV-1 PID. **b** Interaction map of Fab 3D6 and the PID peptide. Interactions were determined by PISA analysis[80] and manual inspection. Hydrogen bonds are shown by dashed lines. HIV-1 PID residue S598 (wild-type Cys598) was not visualized in this structure. HIV-1 residue S604 (labeled Cys604Ser) is coordinated by a hydrogen bond with Fab 3D6 residue S56H. **c** Stick representation of the antigen-binding site in which the HIV-1 PID peptide is colored in the dark green, heavy chain in yellow, and light chain in purple.

structures suggest that the PID motif is present as multiple distinct states on gp41 spikes on the surface of HIV-1 virions.

**Conformational plasticity of the HIV-1 gp41 PID**. The crystallographic data indicated that the HIV-1 gp41 PID peptide exists in at least two conformations when bound to patient-derived Fabs specific to the immunodominant region. In order to better understand the dynamics of the PID region, we utilized MD simulations to characterize its conformational plasticity. First, we performed a 1-μs time scale MD simulation in solution on a 15-residue stretch of the PID region. The end-to-end distance of the N- and C-termini of the PID region was used as a measure of the conformation. We found that the full conformational space was explored within 100 ns, with the PID end-to-end distance ranging from 0.5 to 3.2 nm. Four metastable states are observed with end-to-end distances of 1.2, 1.9, 2.4, and 2.8 nm (Supplementary Fig. 4). The 2.4-nm state was present in 47% of the simulations, whereas the next most stable 1.9-nm state was present in 32% of the simulations. The 2.4-nm and 1.9-nm states are consistent with the end-to-end distances of the PID conformations observed in the bound 3D6 and F240 structures, respectively. Importantly, the observed metastable states were reproducible whether the PID starting model used in the MD simulation was the conformation extracted from 3D6 or F240 complex crystal structures or a linearized version of the PID

motif (Supplementary Fig. 4). This suggests that the PID region adopts an ensemble of conformations in solution. When the PID is bound to Fab F240 or 3D6, it adopts a metastable state with an end-to-end distance of 1.9 or 2.4 nm, respectively (Supplementary Fig. 4).

MD simulations were also performed on a homology modeled HIV-1 gp41 trimeric ectodomain structure in order to understand the dynamics of the PID in the context of the native HIV-1 gp41 protein. Multiple 200-ns MD simulations indicated that there are two regions of flexibility between residues 596–610 and residues 617–623 (Fig. 6, Supplementary Fig. 5, and Supplementary Movie). The residues 596–610 correspond precisely to the HIV-1 gp41 PID motif. In contrast, the central helical heptad-repeat regions (HR1 and HR2) remained structurally rigid during the entire simulation.

The PID C598–C604 disulfide bond is understood to be labile and heterogeneous. Weissenhorn et al.[39] showed that the PID formed both inter- and intra-protomer disulfides and suggested that the disulfides undergo exchange during activation of gp41 to the membrane fusion state. Therefore, we did not apply constraints in the simulation to the PID Cys-Cys disulfide bond. Analysis of our MD simulations over a representative 100 ns snapshot found that the sulfur–sulfur distance (R-SH SH-R') displayed a high level of stability. The positional variability of the sulfur–sulfur distance was 1/10th the variability of the overall Gly-Trp end-to-end position, suggesting the sulfur–sulfur

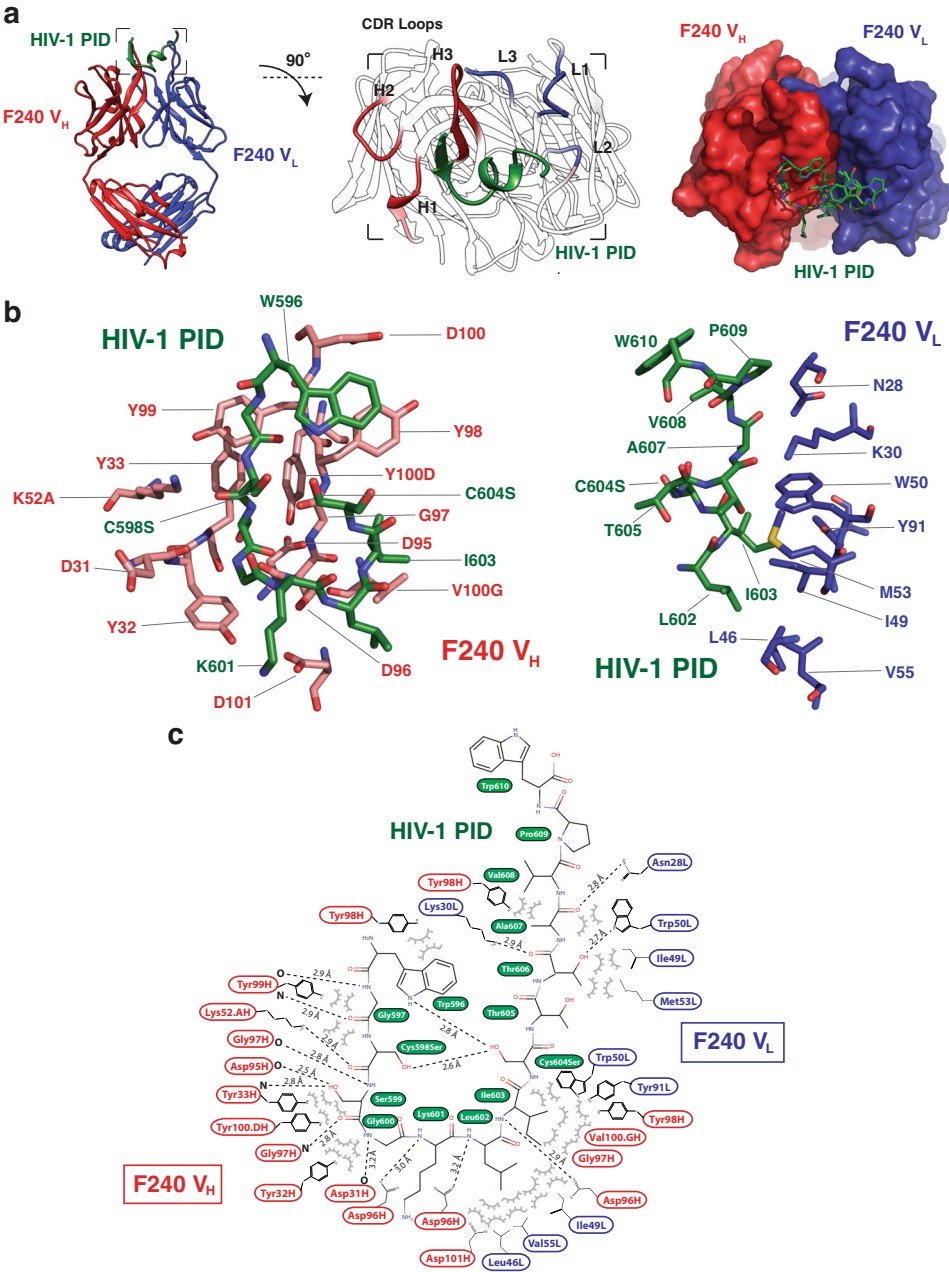

**Fig. 3 Overall structure of Fab F240 complexed to HIV-1 gp41 PID. a** The crystal structure of the Fab F240-PID complex shows the PID region in a β-strand-loop-helix conformation coordinated primarily by CDR loops L1, L2, H1, and H3. **b** Stick representation of the antigen-binding site in which the HIV-1 PID peptide is colored in the dark green, heavy chain in red, and light chain in dark blue. **c** Stick representation and interaction map of Fab F240 and the PID peptide. HIV-1 PID residue S598 (labeled Cys598Ser) and S604 (labeled Cys604Ser) recapitulate a 2.6-Å intramolecular disulfide bond. Interactions were determined by PISA analysis[80] and manual inspection. Hydrogen bonds are shown by dashed lines.

distance was stable without introducing further constraints. Overall, the lack of disulfide constraints likely did not lead to an increase in the observed gp41 flexibility in the simulations.

The flexibility of the PID motif is specific to the post-fusion gp41 conformation. Analysis of a 2-µs simulation of a fully glycosylated pre-fusion HIV-1 SOSIP gp160[40] revealed minimal fluctuations in the PID motif (Fig. 6). The root-mean-square fluctuation (RMSF), a measure of individual residue flexibility or how much a particular residue moves during MD simulations, was fivefold greater in the post-fusion gp41 conformation than in the pre-fusion. Further detailed analysis of the PID motif revealed that this region reliably explored at least four major conformational states in three replicates of 200-ns MD simulations (Fig. 7).

These major conformational states agreed well with MD simulations of the free 15-mer PID motif (Supplementary Fig. 4).

The underlying basis of PID conformational plasticity is likely due to the presence of stretches of amino acids with high flexibility coupled with an alteration of tertiary structure. In pre-fusion gp160, residues 602–606 of the PID form an intermolecular β-sheet with β4 and β26 strands of gp120[37]. Following the conformational change to post-fusion gp41, the PID is thought to be displayed at the apex of the trimeric spike, unencumbered by the gp120 subunit. Without the constraints placed on the PID from gp120, intrinsic characteristics of the primary and secondary structure become dominant. The conformational flexibility of amino acids was previously characterized with the following

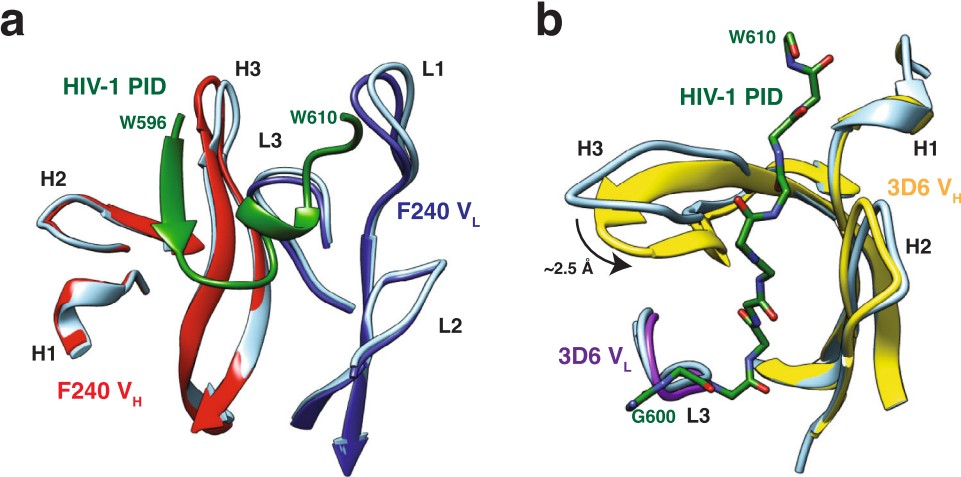

**Fig. 4 Lock-and-key mechanism of interaction of PID with Fab F240. a** Bound and unbound F240 superimpose with an 0.65 Å RMSD with 98.2% of equivalent positions overlapping, suggesting a lock-and-key mechanism of binding. **b** Conformational changes within Fab 3D6 CDR loops upon coordination of the PID peptide. 3D6 unbound and PID-bound structures superimpose with a 0.90 Å RMSD with 98.4% overlap. Ribbon diagrams of unbound Fabs are depicted in cornflower blue, whereas the PID is colored in the dark green, heavy chain in red, and light chain in dark blue.

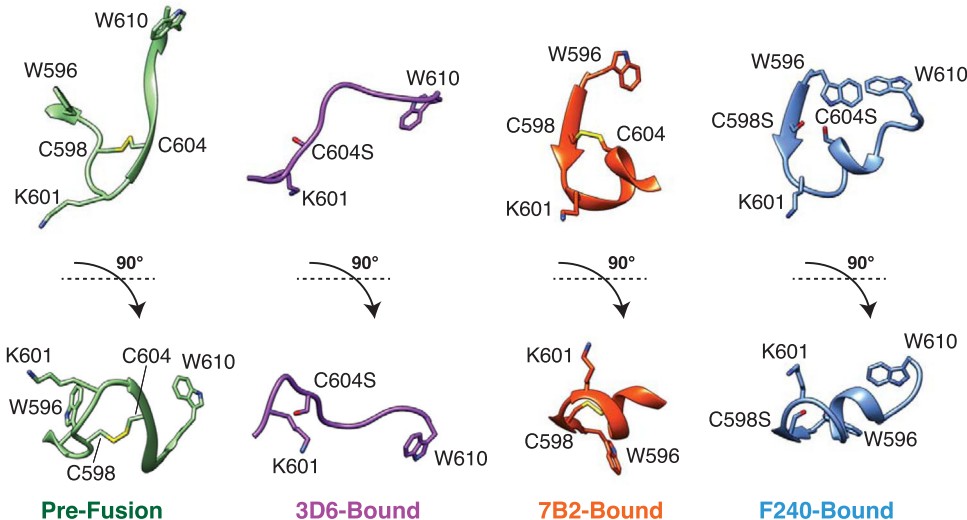

**Fig. 5 Conformations of the HIV-1 gp41 PID motif.** Comparison of the PID region (residues 596–610) in the pre-fusion HIV-1 gp160 (PDB: 6DE7), Fab 3D6-bound, Fab 7B2-bound (PDB: 4YDV), and Fab F240-bound conformations. HIV-1 PID residue S598 (labeled C598S) and S604 (labeled C604S) are shown here in stick representation.

order of flexibility[41]: Gly>Ser>Asp/Asn/Ala>Thr/Leu>Phe/Glu/Gln/His/Arg>Lys>Val>Ile>Pro. In general, residues that are bulky, contain side-chain β-branching, or enable charge repulsion reduces conformational flexibility[41]. Proline is the most rigid, while glycine is not surprisingly the most flexible; residues that favor β-turns also have a high degree of flexibility. Overall, more than half the residues in the PID motif (8/15 residues) are considered conformationally flexible. In addition, there are no long stretches of highly rigid amino acids. The flexible residues in the PID motif are localized to two stretches of amino acids ([596]WGCSGKLICTTAVPW[610]). The first region of flexibility ([597]GCSG[600]) is able to adopt a random coil and β-strand, as observed in the Fab 3D6-bound and F240-bound structures, respectively. The rotational freedom of the [597]GCSG[600] residues allows this region to be malleable to form a β-sheet with CDR-H3 when bound to Fab F240. The second region of flexibility ([604]CTTA[607]) follows a region ([601]KLI[603]) that strongly favors the formation of an α-helix. The [601]KLI[603] region is only able to form a short helix; the presence of threonine residues in

[604]CTTA[607] tends to disfavor helix formation due to β-branching of its side chain restricting the range of rotation. We hypothesize the [604]CTTA[607] region destabilizes the PID α-helix to allow the PID to adopt multiple conformations. Alignment of HIV-1 gp41 PID sequences representing the diversity in the global HIV-1 pandemic illustrates the conserved nature of the PID region (Fig. 1), thus this flexibility observed in MD simulations is likely a phenomenon that occurs in most HIV-1 isolates. We conclude that the PID is flexible but can form an ensemble of conformations capable of being recognized by various non-neutralizing antibodies, thereby facilitating HIV-1 immunodominance observed in acute and chronic HIV-1 infections.

## Discussion

Despite a search for a protective HIV-1 vaccine that has spanned over 30 years, there has been no success to date. Eliciting broadly neutralizing antibodies is considered crucial in generating a protective immune response and an effective vaccine. The

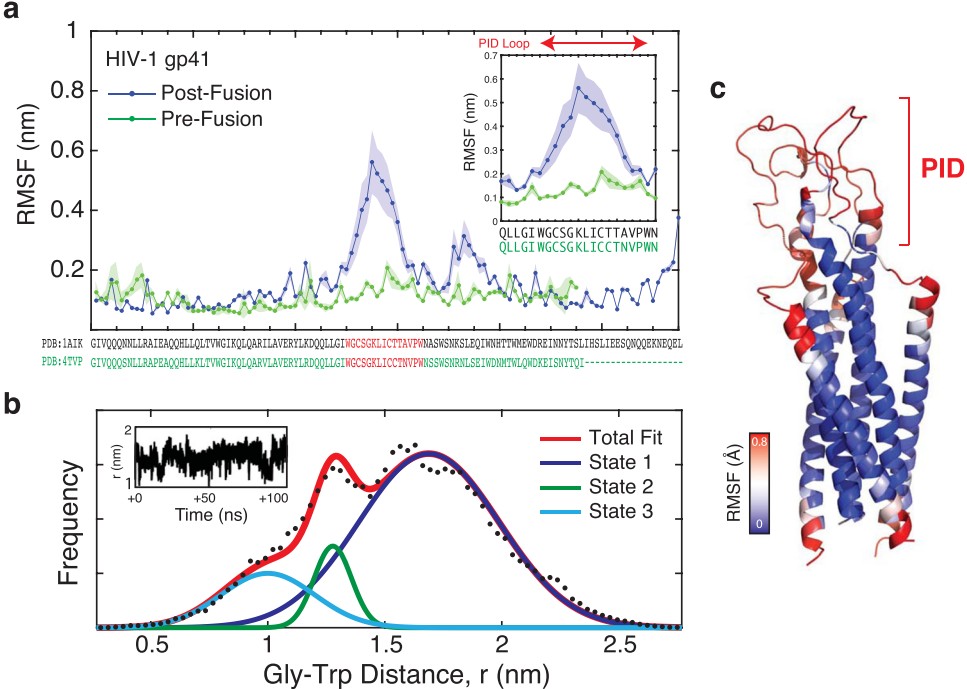

**Fig. 6 Conformational plasticity of the HIV-1 gp41 ectodomain. a** MD simulation of a homology model of the complete HIV-1 gp41 ectodomain. RMSF plot versus residue show flexibility of the PID region in the HIV-1 gp41 post-fusion state, but not the pre-fusion conformation. The inset panel shows an expanded view of the RMSF structural fluctuations in the PID region between the pre- and post-fusion conformations. **b** Analysis of the MD simulation of the PID region in the HIV-1 gp41 ectodomain. Raw data showing the frequency of end-to-end distance measurements between Gly600 and Trp610 of the PID region (black dots) are used to calculate an overall total fit (red curve). Potential metastable states are deconvoluted from the total fit, as estimated by distinct Gaussian distributions. Upon fitting, three metastable conformations are observed, with the most populated state centered at 1.8 nm (State 1) and two additional minor states at 1.3 nm and 1.0 nm (State 2 and State 3). This suggests that the PID region adopts a distribution of conformations in simulation. **c** Structure of the homology model of the HIV-1 gp41 ectodomain superimposed with a heat map of RMSF displacements (in Å) over the course of the MD simulation. Regions of the highest flexibility are highlighted in red.

generation of broadly neutralizing antibodies is possible but rare[42–44]. Considerable efforts have been made to understand the antibody evasion strategies of HIV-1. Structural features of the HIV-1 envelope glycoprotein, gp160, are important for the virus' ability to escape antibody surveillance as are sequence variation[45,46], shielding of gp160 surfaces by heterogenous glycans[47], conformational masking[48,49], steric restriction[50], protection of conserved surfaces[51–53], low gp160 density that reduces antibody avidity[54], and lack of germline genes capable of recognizing and maturing into broadly neutralizing antibodies[55].

Whereas HIV-1 gp160 has been the focus of intense interest given its importance as a vaccine target, this is not the only virally encoded protein on the surface of HIV-1 that is important in immune surveillance. Shedding of gp120 leads to gp41 stumps that cover the viral membrane. The gp41 adopts a post-fusion conformation that is non-functional but contains immunodominant epitopes recognized by antibodies. In general, the presence of immunodominant epitopes correlates with high levels of viral clearance and control. For example, antibodies against hepatitis C virus (HCV) genotype 1a immunodominant epitopes have been associated with a lower viral load in infected patients[56], and immunodominant antibodies from the IGHV1–69 locus are directed against the hemagglutinin stem region of influenza A virus[57]. Preferential usage of the IGHV1–69 to produce broadly neutralizing antibodies is well described at the population level[58]. Interestingly, immunodominant antibodies from this same locus are upregulated during acute HIV-1 infection, but the antibodies produced against HIV-1 gp41 are non-neutralizing[59]. PID-driven immunodominance does not result in potent neutralizing antibodies likely

because gp41 is displayed on the surface of the virus in its post-fusion triggered state.

Studies of several HIV Vaccine Trials Network clinical trials indicate that dominant non-neutralizing antibody responses against HIV-1 gp41 (but not the PID) were primed by the host microbiome. This was determined by cross-reactivity and genetic lineage analyses[59,60]. Our structures and MD simulations on gp41 now propose another mechanism that HIV employs to evade antibody surveillance. We hypothesize that structural plasticity contributes to a different mechanism of immune evasion.

The conformational flexibility of antigens plays a major role in antibody recognition and immune responses, but conformational plasticity generally leads to the reduction of antigen immunogenicity[61–64]. For example, in HIV-1, the CD4 binding site within its gp120 attachment subunit is structurally disordered and does not elicit a strong neutralizing antibody response[62]. In HCV, the CD81 binding loops have large conformational flexibility that prevents antibodies from being generated to this site[63,64]. Motions of surface glycoproteins prevent tight antibody binding in part due to an entropic penalty. It has been hypothesized that viruses with highly disordered virion surface proteins, such as HIV-1, hepatitis C virus, and herpes simplex virus, are less vulnerable to antibody detection and neutralization[61]. For vaccine design, this has important implications. For example, once a respiratory syncytial virus F epitope was stabilized on a scaffold protein, the construct elicited potent neutralizing antibodies[65].

Our data suggest that conformational plasticity of the HIV-1 gp41 PID does not reduce antibody recognition but rather presents an ensemble of surfaces that allows for multiple antibodies

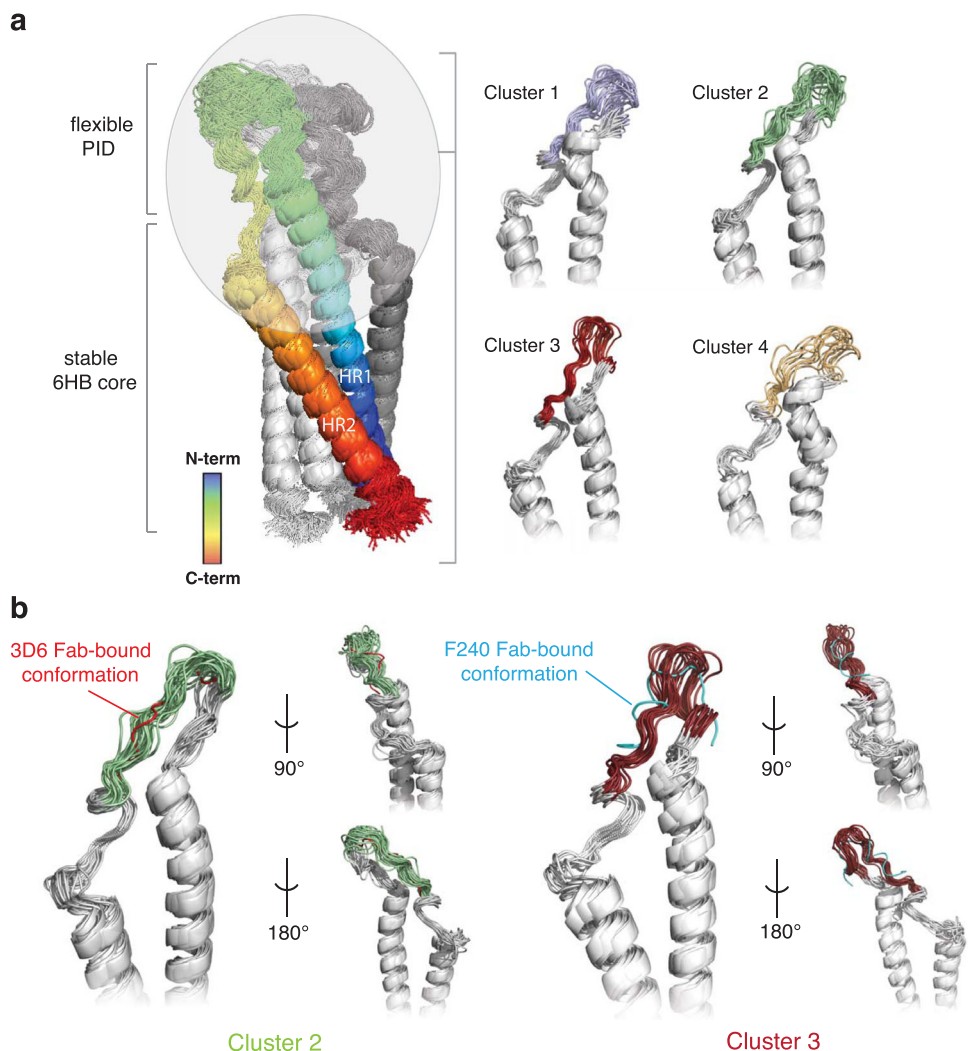

**Fig. 7 Cluster analysis of the HIV-1 gp41 MD simulations. a** Conformational flexibility of PID motif in the context of the full-length gp41 ectodomain based on cluster analysis of a 200-ns MD simulation. The cartoon is colored as a chainbow from blue (N-terminus) to red (C-terminus). The conformations sampled by unbound PID were clustered using backbone RMSD-based hierarchical average linkage clustering with a merging distance cutoff of 2 Å over the trajectory. Based on this cutoff, a total of four clusters were generated, with conformations in each cluster superposed. **b** The gp41 PID conformations observed in clusters 2 and 3 from one representative replicate are consistent with those observed in the Fab 3D6 (red) and F240 (cyan) complex structures.

to bind. Based on the crystal structures of PID bound to Fabs and MD simulations of the complete trimeric gp41 ectodomain, the PID motif has at least two conformations each capable of robustly binding different non-neutralizing antibodies. Fab 3D6 illustrates that at least one of the HIV-1 PID conformations can be readily engaged by near-germline antibodies. This suggests that the PID acts as an immunodominant decoy that focuses the immune reaction toward the irrelevant HIV-1 gp41 stumps.

According to the "doctrine of original antigenic sin" proposed by Thomas Francis, Jr.[66,67], at first exposure, the adaptive immune system mounts and traps a response against the dominant epitopes of the antigen. This immunodominance depends on the competition for helper T cells, which is a limiting factor in the early stages of the immune response. An antigen that binds with high affinity to the BCR will be more robustly presented to helper T cells than weaker binding competitors. Insufficient T-cell stimulation leads to the suppression of B-cell clones, thus trapping the first response to immunodominant motifs. The extreme variability of epitopes on HIV-1 gp120 and gp160 dilutes the activation of the immune response toward these sites that would result in neutralization, whereas conserved elements, such as the

highly conserved PID motif, become the immunodominant motifs. As physiochemical features of an antigen and its interactions with naive BCR govern immunogenicity[68], the selection of an epitope is due to the thermodynamics of binding. The HIV-1 PID region adopts multiple structures capable of interacting with both near-germline and more affinity-matured antibodies. Thus, our data support the hypothesis that the ability of the HIV-1 PID to bind to naive germline-like BCR results in an initial immune response to gp41 and subsequent rapid proliferation of ineffective non-neutralizing antibodies. This mechanism may hinder the formation of broadly neutralizing antibodies in HIV-1-infected individuals.

## Conclusions

The PID region is a highly conserved amphipathic motif in the mutable HIV-1 genome, supporting its important role in HIV pathology. Using a combination of structural biology and computational MD simulations, we have conceptually broadened our understanding of the structural basis of PID-driven immunodominance. We showed that the consensus sequence of the PID

from HIV-1 M group subtype B, which is prevalent in North America and Europe, adopts an ensemble of conformations in the unbound state. Although antigen conformational flexibility generally leads to poorer recognition and binding of antibodies, in the case of gp41 PID, the formation of an ensemble of gp41 conformations allows the binding of both near-germline and affinity-matured patient-derived antibodies. The interactions with near-naive antibodies suggest that the PID region interacts with the germline BCR, resulting in preferred clonal selection and expansion. This immunodominance may hamper the development of productive immune responses to neutralizing hotspots on gp160, thus enabling HIV-1 to establish a chronic and latent infection. Other viral pathogens may use the same strategy of expressing conformationally flexible glycoproteins on the virion surface to act as immunological decoys to prevent elicitation of antibodies against epitopes with more neutralizing potential. The general principles presented here may be useful in efforts to develop HIV-1 vaccine candidates. Stabilization of the PID motif to decrease the disorder and large-scale motions of this region may shift the adaptive immune system to target other sites.

## Methods

**Expression and purification of Fab 3D6 and F240.** The coding sequences for the immunoglobulin light chain and heavy chain variable regions of F240 and 3D6 were previously reported[18,69]. For expression, constructs corresponding to the heavy and light chains were designed and generated (Supplementary Figs. 6 and 7). The Fab F240-$V_H$ and Fab 3D6-$V_H$ constructs include the immunoglobulin heavy chain variable region of F240 or 3D6, respectively, fused to the Igγ-1 chain constant region (Uniprot Accession: P01857) followed by a thrombin protease recognition sequence, a GSSG linker, and a C-terminal deca-histidine tag to facilitate purification. Fab F240-$V_L$ and Fab 3D6-$V_L$ include the immunoglobulin light chain variable region of F240 or 3D6, respectively, fused to the IgK chain constant region (Uniprot Accession: P01834). DNA sequences corresponding to these two constructs were codon-optimized, commercially synthesized (Integrated DNA Technologies), and subcloned into separate custom pcDNA3.4 vectors (Invitrogen) by ligation-independent cloning.

Fab F240 and 3D6 were expressed as secreted proteins from HEK293T cells (ATCC; cat no. CRL-3216) in large-scale cultures[70]. In brief, $2 \times 10^8$ HEK293T cells maintained in Dulbecco's mixed eagle medium (DMEM) supplemented with 10% (v/v) heat-inactivated fetal bovine serum (FBS; Gibco) were seeded into a 6400-cm$^2$ CellSTACK (Corning) containing 1.2 L DMEM with 5% (v/v) FBS and incubated at 37 °C and 5% $CO_2$. After 24 hours, 420 µg pcDNA3.4-Fab F240-$V_H$ or pcDNA3.4-Fab 3D6-$V_H$ and 420 µg pcDNA3.4-Fab F240-$V_L$ or pcDNA3.4-Fab 3D6-$V_L$ were transiently transfected into the HEK293T cells using a 6:1:1 mass ratio of linear 25-kDa polyethyleneimine (Polysciences Inc.) to the plasmids. The transiently transfected cells were incubated for 4 days at 37 °C and 5% $CO_2$, and then the conditioned media was harvested. An additional 1.2 L of DMEM with 5% (v/v) FBS was added to the cells. After incubation for another 4 days at 37 °C and 5% $CO_2$, cells were harvested. This cycle was repeated four times before protein yield dropped below 1 mg L$^{-1}$. Following each harvest, the conditioned media was concentrated by tangential flow filtration using a 10-kDa MWCO membrane (Pall). During concentration, the media was buffer-exchanged by dilution into 20 mM Tris-HCl, pH 9.0, 500 mM NaCl, and 70 mM imidazole. The Fab in the concentrate was then purified by standard Ni-NTA affinity chromatography and concentrated to >1 mg mL$^{-1}$ for storage at 4 °C. Prior to crystallization trials, Fab F240 and 3D6 were further purified by SEC using an ENrich SEC 650 10 × 300 gel filtration column (Bio-Rad) equilibrated in 20 mM Tris-HCl, pH 8.5, and 150 mM NaCl. Protein concentration was determined based on absorbance at 280 nm, and purity was monitored by sodium dodecyl sulfate–polyacrylamide gel electrophoresis and mass spectrometry.

**Characterization of F240 and 3D6 binding to HIV-1 gp41.** Absorbance ELISA using the HIV-1 Consensus Subtype C Env Peptide Set (NIH AIDS Reagent Program; cat. No. 9499) was performed as previously described[71] with minor modifications. 15-mer peptides comprising the HIV-1 gp41 PID were solubilized in 100 mM sodium bicarbonate, pH 9.3, diluted to 10 µg mL$^{-1}$, and adsorbed to Costar 96-well half-area, flat-bottom microtiter plates (Corning) by incubating 50 µL volumes overnight at 4 °C. Following coating, the microtiter plates were washed five times with PBST (1× PBS, 0.1% (v/v) Tween-20) and blocked with 150 µL of PBST-Milk (PBST supplemented with 5% (w/v) skim milk powder (BioShop)) for 2 hours at room temperature. After decanting the blocking reagent and washing three times with PBST, 50 µL volumes of primary anti-gp41 Fab F240 or 3D6 at 1 µg mL$^{-1}$ in PBST-Milk were incubated with the adsorbed HIV-1 gp41 peptides for 1 hour at room temperature. The plates were then washed five times with PBST, and 50 µL of secondary goat anti-human horseradish peroxidase (HRP)

conjugated antibodies (1:20,000 dilution; Thermo Pierce) in PBST-Milk was incubated for 1 hour at room temperature. Following 10 washes with PBST, the ELISA plates were developed using 50 µL of TMB-One HRP substrate (Kem-En-Tec). Color development was fixed after 2 minutes by the addition of 50 µl of 2 N sulfuric acid. Color development was promptly read at 450 nm on a Tecan Infinite microplate reader.

**Crystallization of apo-Fab F240 and 3D6.** Initial sparse-matrix crystallization screening of Fab F240 (12 mg mL$^{-1}$) and Fab 3D6 (15 mg mL$^{-1}$) was performed by sitting-drop vapor diffusion in 96-well two-drop UV Intelliplates (Art Robbins) using the Douglas Instruments Oryx 8 liquid handling system. Fab F240 crystals were obtained in 0.1 M HEPES-NaOH, pH 7.5, and 20% (w/v) PEG 8000 at 293 K. Fab 3D6 crystals were obtained in 0.17 M ammonium sulfate, 25.5% (w/v) PEG 4000, and 15% (v/v) glycerol. Cryoprotection of Fab F240 was achieved by sequential soaking of crystals in mother liquor containing 5–25% (w/v) glucose, and crystals were flash cooled in liquid $N_2$. Cryoprotection of Fab 3D6 crystals was achieved by transferring the crystals to perfluoropolyether oil (Hampton Research) prior to flash cooling in liquid $N_2$.

**Crystallization of Fab F240-gp41 PID and 3D6-gp41 PID peptide complexes.** HIV-1 gp41 PID peptide was commercially synthesized (Biomatik; Kitchener, Canada) with cysteine to serine double mutations (C603S and C609S) to avoid the formation of intermolecular disulfide bonds (acetyl-WGCSGKLICTTAVPW-amidated). Prior to crystallization trials, the peptide was dissolved in 20 mM Tris-HCl, 150 mM NaCl, pH 8.5 and purified by isocratic elution in 20 mM Tris-HCl, pH 8.5, and 150 mM NaCl on an ENrich SEC 70 10 × 300 gel filtration column. Fab F240 and Fab 3D6 were complexed with HIV-1 PID peptides by combining purified Fabs with SEC-purified peptides at a 1:5 molar ratio and incubated overnight at 4 °C. Sparse-matrix crystallization screening of Fab F240-PID complex (14 mg mL$^{-1}$) and Fab 3D6-PID complex (15 mg mL$^{-1}$) was performed as described above. Crystals of the Fab F240-PID complex formed in 0.17 M ammonium acetate, 0.085 M sodium citrate-HCl, pH 5.6, 25.5% (w/v) PEG 4000, and 15% (v/v) glycerol. For Fab 3D6, spherulites and small bundles of rod-like crystals were obtained after several days of incubation in mother liquor containing 0.2 M magnesium chloride, 20% (w/v) PEG 8000, and 0.1 M Tris-HCl, pH 8.5. These crystals were improved via additive screening using the Detergent Screen HT (Hampton Research). The addition of 0.59 mM n-undecyl-$\beta$-D-maltoside to the sitting-drop formulation promoted the growth of singular crystals suitable for diffraction. Cryoprotection of both Fab F240-PID and Fab 3D6-PID complexed crystals was achieved using perfluoropolyether oil (Hampton Research) prior to flash cooling in liquid $N_2$.

**Data collection.** All crystals were diffracted remotely on beamline 08ID-1 at the Canadian Light Source (Saskatoon, SK). All data sets were reduced using XDS and scaled using the program Aimless[72] from the CCP4 suite. Data quality was assessed using both $CC_{1/2}$ and standard conventions such as $R_{merge}$ and I/σ(I). Following data processing, the data from the Fab F240-PID complex crystal were transformed from $P2_122_1$ to $P2_12_12$ using the CCP4 program REINDEX[73].

**Structure determination of apo-Fab F240 and F240-PID complex.** Structural determination of apo-Fab F240 was accomplished by molecular replacement using the program PHENIX.phaser[74] and an edited structure of Fab 3D6 (PDB: 1DFB) as a search model. Fab 3D6 was separated into the variable (residues H1–H103, H114–H122, L1–L23, and L34-L90) and constant domains (residues H130–H229 and L108–L212). Additionally, any Fab 3D6 residues that did not match the sequence of Fab F240 were truncated to alanine in the coordinate file prior to molecular replacement. The top molecular replacement translation function $Z$ score was 22.8, indicating a correct solution.

Following molecular replacement, PHENIX.autobuild was used to build sidechains and extend the placed search model. Iterative rounds of manual model rebuilding and refinement were performed using the programs Coot[75] and PHENIX.refine[76], respectively. Torsion-angle simulated annealing refinement (5000 K starting temperature) with individual atomic displacement and Translation/Liberation/Screw groups was carried out. Of unique reflections, 5% were reserved during refinement as the test-set. Annealed $2|F_o|-|F_c|$ composite omit maps were used to minimize model bias during rebuilding. Clear electron density was seen for residues H1–H218 and L2–L214 in Fab F240; however, no main-chain electron density was observed for residues H127–H134, within the heavy chain constant region.

The Fab F240-PID structure was determined by molecular replacement using the refined structure of Fab F240 as a search model in PHENIX.phaser. The top molecular replacement translation function $Z$ score was 54.3, indicating a correct solution. The model was rebuilt using PHENIX.autobuild and then subjected to iterative rounds of manual model rebuilding and refinement as described above. Clear electron density was seen for the entirety of the PID, whereas clear electron density was observed for residues H1–H213 and L3–L212 in Fab F240. Much like the apo-structure, no main-chain electron density was observed for residues H127–H134 in the Fab F240-PID complex.

**Structure determination of apo-Fab 3D6 and 3D6-PID complex**. Structural determination of apo-Fab 3D6 was accomplished by molecular replacement using the program PHENIX.phaser[74] and the same search model utilized above for the determination of the apo-Fab F240 structure. The top molecular replacement translation function $Z$ score was 38.3, indicating a correct solution. Identical procedures to those for apo-Fab F240 were employed to build the apo-Fab 3D6 model. The structure was refined using PHENIX.refine[76] and PDB-REDO[77].

The structure of Fab 3D6 in complex with the PID peptide was determined by molecular replacement using the refined structure of Fab 3D6 as a search model in PHENIX.phaser. The CDR-H3 loop was truncated between $V_H$ residues 97 and 100 to accommodate possible conformational changes induced upon epitope binding. The top molecular replacement translation function $Z$ score was 76.7, indicating a correct solution. Clear electron density was observed for much of the PID peptide. The peptide was built as a new chain, and the model was rebuilt using alternating rounds of PHENIX.autobuild. The resulting model was then subjected to iterative rounds of manual model rebuilding and refinement as previously described. NCS restraints were employed throughout the refinement.

For all structures, the stereochemical quality of the refined models was validated using MolProbity[78] and Coot[75]. The quality of $R_{work}/R_{free}$, B-factors, and root-mean-square deviations (RMSDs) of bond lengths and angles of all structures are consistent with other deposited structures determined at similar resolutions, as validated by PHENIX.polygon[79]. Antibody fragment sequences are labeled according to Kabat convention. Fab-peptide interactions were identified and analyzed using the PDBePISA server[80]. All data collection and refinement statistics are presented in Table 1.

**Preparation of structures for MD simulation**. MD simulations were performed to understand the conformational flexibility of the HIV-1 gp41 PID motif. For the full-length ectodomain analysis of HIV-1 gp41, a homology model of the complete gp41 ectodomain (HXB2 isolate; Uniprot accession #P04578; residues 2–116) was generated using Phyre2[81]. A post-fusion HIV-1 gp41 ectodomain was generated by trimerizing the monomers using the program Coot and the SIV gp41 trimer as a guide (PDB code: 2EZO). The final HIV-1 gp41 homology model was energy minimized prior to MD simulations. For the free PID peptides, the coordinates corresponding to the PID region were extracted from the Fab-PID structure and used directly in conformation bound to the Fab or linearized simulations. For Fab-PID complexes, coordinates for a single Fab-PID complex were extracted from the crystallographic structures. There are no glycosylation sites on the PID region in gp41, thus no carbohydrates were modeled onto the structures. Hydrogen atoms and bond orders were assigned using the default protocol from CHARMM-GUI in agreement with the CHARMM36m force-field.

**MD simulation**. All PID peptides (either in a conformation bound to the Fab or linearized), Fab-bound PID complexes, and the HIV-1 gp41 trimer were placed in the center of a box with an edge distance of 1 nm applied in all directions from the protein. The box was then populated with water molecules and KCl ions to a final concentration of 150 mM with the CHARMM-GUI web server[82]. No constraints were applied to the PID disulfide bond (C598–C604) in the simulations. Parameters for the MD simulations are presented in Supplementary Table 1.

All simulations were performed using the GROMACS 5.1.4 software package, utilizing the CHARMM36m force field[82,83]. All simulations used a 2-fs time step, a periodic boundary condition applied in all directions, a short-range van der Waals cutoff of 1.2 nm using a Verlet cutoff scheme, the particle-mesh Ewald solution for long-range electrostatics, and the P-LINCS algorithm for the determination of bond constraints. Water was modeled using the TIP3P model. A Nose-Hoover thermostat at 310 K was used for temperature coupling at a time constant ($\tau_t$) of 0.5 ps, and a Parrinello-Rahman semi-isotropic weak pressure coupling scheme was used to maintain a pressure of 1.0 bar with $\tau_p$ of 1 ps. The systems were first energy minimized with the steepest descent method until a convergence threshold of $1000\ \text{kJ mol}^{-1}\ \text{nm}^{-1}$ was reached. This was followed by a 1-ns simulation with an NVT/NPT ensemble (constant number of particles, pressure, and temperature), and the convergence of energies, temperature, pressure, and density of the systems were reached. Unrestrained MD simulations and analyses were conducted as described in Supplementary Table 1 with adequate time given for protein equilibration. MD simulations were carried out using the Compute Canada Graham supercomputer.

**MD trajectory and cluster analysis**. The simulations were analyzed with standard GROMACS scripts and MATLAB 2019b with the SpectralTools and Mfit4 plug-ins. Analysis of trajectories was performed by calculating an RMSF from the reference structure over the course of the simulation. RMSF is defined as:

$$RMSF = \sqrt{\langle |p(t) - p_0|^2 \rangle},$$ where $p$ is the center of mass position of each residue at time $t$ compared to the reference structure $p_0$. To understand whether discrete structures are stable over the course of the simulation, the end-to-end distance ($d$) between the Cα center of masses of the first and fifteenth residues of the PID motif was measured. End-to-end distance was binned and fit with a series of Gaussian distributions with least squares, fit parameters were then extracted for analysis. Geometric clustering was done with the *gmx cluster* function and the gromos algorithm with a merging distance cutoff of 2 Å[84]. This was conducted with each

100-ns trajectory split into 100 frames, and the proportion of time each protein remained was determined as the stability. A cutoff of 10% was used to determine whether particular structures were metastable. The cluster versus stability ratios is provided in Supplementary Fig. 5. Visualization and all molecular representations were generated with PyMOL and VMD 1.9.3.

**Statistics and reproducibility**. The results are presented as mean ± standard deviation (SD). Technical triplicates were performed for all ELISA-binding assays. For MD simulations, $n = 3$ independent MD replicas were performed. The reproducibility of the HIV-1 gp41 ectodomain MD simulations is demonstrated by the similarity of the RMSF plot, cluster stability, and PID end-to-end distance frequency of the three replicates in Supplementary Fig. 5.

**Reporting summary**. Further information on research design is available in the Nature Research Reporting Summary linked to this article.

## Data availability

Atomic coordinates and structure factors for patient-derived Fabs with and without the HIV-1 gp41 PID peptide have been deposited in the Protein Data Bank with the accession codes as follows: apo-Fab 3D6: 7N07; Fab 3D6-PID complex: 7N08; apo-Fab F240: 7N04; Fab F240-PID complex: 7N05. Further information and reasonable requests for resources and reagents should be directed to the Corresponding Author, Jeffrey E. Lee (jeff.lee@utoronto.ca).

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

## Acknowledgements

This work was supported by funding from the Canadian Institutes of Health Research (MOP-115066 and PJT-173301), Ontario Early Researcher Award (ER-13-09-116), and Canada Research Chair in Structural Virology (950-231672) to J.E.L. Biophysics and structural biology infrastructure were supported by funding from the Canada Foundation for Innovation John R. Evans Leaders Fund (CFI-JELF). Support for stipends was provided by a Vanier Canada Graduate Scholarship and Ontario Graduate Scholarship to J.D.C. This work is based upon remote data collection on beamline 08ID-1 at the Canadian Light Source (CLS). The CLS is supported by the Canada Foundation for Innovation, Natural Science and Engineering Research Council of Canada (NSERC), National Research Council of Canada, CIHR, the Province of Saskatchewan, Western Economic Diversification Canada, and the University of Saskatchewan. MD simulation research was enabled in part by support provided by Compute Ontario (www.computeontario.ca) and Compute Canada (www.computecanada.ca). The following reagent was obtained through the AIDS Reagent Program, Division of AIDS, NIAID, NIH: HIV-1 Consensus Subtype C Env Peptide Set. The authors thank Drs. Thomas Lemmin and Peter Kwong for providing the trajectories of pre-fusion HIV-1 SOSIP trimer.

## Author contributions

J.D.C. and J.E.L. conceptualized and designed the research. J.D.C. produced the antibody fragments, conducted the ELISA peptide binding assays, and determined and analyzed the crystal structures. A.K. performed and analyzed the MD simulations. J.D.C., A.K., and J.E.L. wrote the manuscript. J.E.L. acquired funds and supervised the research. All authors discussed the results and approved the manuscript.

## Competing interests

The authors declare no competing interests.
