## [Peer Review File · Communications Biology]

Reviewers' comments:

Reviewer #1 (Remarks to the Author):

"Conformational plasticity of the HIV-1 gp41 immunodominant region is recognized by multiple non-neutralizing antibodies", by Cook et al.

In this manuscript, the authors use a combination of X-ray crystallography, MD simulations, and epitope mapping to demonstrate a conformationally flexible PID. This is suggested to be important for mis-direction of the immune response.

Overall, this is a well-written manuscript. I only have minor comments. I support publication after these modifications have been made.

Line 43: Missing reference to the RV144 trial

Line 69: "Antibodies" suggests a divalency/multi-valency. I suggest changing "antibodies" to Fabs for accuracy.

Line 107: "In our unbound Fab 3D6 structure, CDR-H3 is mostly disordered, suggesting some intrinsic flexibility". Is there a sequence/pattern in this loop that favors disorder?

Line 202: "Conformational plasticity of the HIV-1 gp41 PID ". This is extremely interesting. However, the authors do not provide a clear explanation of the underlying basis for this peptide flexibility. For instance, is this due to the short length of the peptide, or perhaps the presence of residues that disrupt helices, promote turns, or maybe the lack of intra-chain H-bonding? I suggest probing the results of the MD simulations further to figure out the probable source of this conformational flexibility. This would add to the manuscript.

General question related to line 107: Is this disorder in CDR-H3 involved in any way in broad selectivity? If this loops adopts several conformations, does it present disparate surfaces for interaction with a variety of epitopes? By corollary, would a structured CDR loop be more selective? It would be good if the authors could comment on this briefly.

Reviewer #2 (Remarks to the Author):

Cook and colleagues report the structural characterization of non-neutralizing antibodies targeting the PID on gp41. Their work contributes to the understanding of the PID misdirecting the humoral immune response at early and later stages of HIV-1 infection. Their work shows that the PID region is conformational flexible by revealing different conformations of PID by X-ray crystallography and MD simulation alone or in complex with the close to germ-line like Ab 3D6 and by F240 that carries a higher somatic mutation rate.

The paper is well written, technically sound and provides further insight into the conformational landscape of PID.

The following points need to be addressed to improve the manuscript:

Is there anything known about the infection state at the time of Ab isolation? Was 3D6 isolated at an early stage during infection and F240 at a later stage, explaining the difference in somatic mutations.

Line 120: please add citations to the statement:wild-type post-fusion HIV-1 gp41 via a 2.6Å hydrogen bond between S598 and S604? Furthermore, the residues should be cysteines?

Please highlight the disulfide bond (the ser positions) in the two Fab peptide structures.

Throughout the manuscript, the post fusion conformation of gp41 is described as "broken gp41 spikes " or with similar terms. This should be explained and changed in the text. Consensus is that shedding of gp120 from native trimers induces the post fusion conformation of gp41 that produces the hairpin structure with both TM and Fp inserted in the same membrane, exposing PID at the tip of the rod. Appropriate citations of the gp41 post fusion conformation should be added.

Were the MD simulations performed constraining the PID disulfide bond? If not, this could certainly contribute to the observed flexibility. Note that the post fusion conformation was suggested to exchange disulfides between protomers in the post fusion conformation (Weissenhorn et al. EMBO J 1997) in agreement with the role of disulfide exchanges during entry (Stantchev et al. Retrovirology. 2012).

Reviewer #3 (Remarks to the Author):

This is a well written manuscript with solid data that details investigations of the most immunogenic site on HIV envelope. As this immunologic site is so immunogenic, the relative paucity of knowledge of its antigenic character is surprising, so this paper adds significant knowledge to the field.

The differential presentation of this epitope however is not completely novel. Since its description, this region is known to represent more linear and conformational epitopes (Ref 6- Xu et al, J Virol. 1991 Sep; 65(9):4832-8. doi: 10.1128/JVI.65.9.4832-4838.1991.), was explored in Ref 12 Gohain, and recently shown so in monoclonal antibody discovery (Hicar et al, PLoS One. 2016 Jul 13; 11(7):e0158861. doi: 10.1371/journal.pone.0158861. eCollection 2016.).

Specific comments:

Lines 53-57 is both a bit too definitive and can use more explanation for the general audience. The PID is not completely inaccessible, as F240 in particular shows binding to SOSIP pre-fusion complexes (Kong et al, Nat Commun . 2016 Jun 28; 7:12040. doi: 10.1038/ncomms12040.). Line 56 I think we need another sentence to explain that a significant portion of envelope on the virion surface is stubs or 'broken' env. Further explanation of "broken gp41 spikes" would clarify this as well.

RMSD and RMSF may need to be explained and distinguished better up front. Line 231 used RMSD but figure 4 RMSF.

Point-by-point Response to Reviewer Comments- COMMSBIO-21-2615

“Conformational plasticity of the HIV-1 gp41 immunodominant region is recognized by multiple non-neutralizing antibodies”

JD Cook, A Khondker, and JE Lee

Reviewer #1:

"Conformational plasticity of the HIV-1 gp41 immunodominant region is recognized by multiple non-neutralizing antibodies", by Cook et al.

In this manuscript, the authors use a combination of X-ray crystallography, MD simulations, and epitope mapping to demonstrate a conformationally flexible PID. This is suggested to be important for mis-direction of the immune response.

Overall, this is a well-written manuscript. I only have minor comments. I support publication after these modifications have been made.

1) Line 43: Missing reference to the RV144 trial

Response: Reference has now been added to the sentence.

2) Line 69: "Antibodies" suggests a divalency/multi-valency. I suggest changing "antibodies" to Fabs for accuracy.

Response: Reviewer #1 makes an excellent point. Where appropriate, we have changed the text throughout the manuscript to 'Fabs' for accuracy.

3) Line 107: "In our unbound Fab 3D6 structure, CDR-H3 is mostly disordered, suggesting some intrinsic flexibility". Is there a sequence/pattern in this loop that favors disorder?

Response: We have performed an analysis of the primary sequence of 3D6 CDR-H3 using the program DEPICTER (Zhang *et al. JMB*, 2020, 432, 3379-3387). The CDR-H3 is not predicted to be intrinsically disordered, however there is often intrinsic flexibility in CDR-H3 paratopes of germline and near-germline antibodies. We have now added an updated discussion in the manuscript (lines 111-117).

4) Line 202: "Conformational plasticity of the HIV-1 gp41 PID ". This is extremely interesting. However, the authors do not provide a clear explanation of the underlying basis for this peptide flexibility. For instance, is this due to the short length of the peptide, or perhaps the presence of residues that disrupt helices, promote turns, or maybe the lack of intra-chain H-bonding? I suggest probing the results of the MD simulations further to figure out the probable source of this conformational flexibility. This would add to the manuscript.

Response: Thank you for Reviewer #1's comment, we also find the conformational plasticity to be very interesting. We have now investigated the nature of the PID residues more carefully to provide a better explanation of the underlying basis for peptide flexibility. The underlying basis of PID conformational plasticity is likely due to the presence of stretches of amino acids with high flexibility. The conformational flexibility of amino acids was previously characterized with the following order of flexibility (Huang *et al. Angewandte Chemi* 2003 42, 2269-72): Gly > Ser > Asp/Asn/Ala > Thr/Leu > Phe/Glu/Gln > His/Arg > Lys > Val > Ile > Pro. In general: 1) large residues reduce flexibility, 2) β -branching increases the activation barrier for amino acid bond rotation thus decreasing conformational flexibility, 3) charge repulsions between residues slightly decrease flexibility, 4) Pro is the least flexible while Gly is most flexible, and 5) amino acids with high β turn propensity has higher conformational flexibility.

The PID region (WGCSGKLICTTAVPW) contains several stretches of highly conformationally flexible residues (GCSG) and moderately conformationally flexible residues (CTTA). Approximately half the residues (8/15 residues) are considered conformationally flexible. Moreover, there are no long stretches of highly rigid amino acids in the PID motif; the longest is a stretch of two amino acids. The flexible residues in the PID motif are localized to two stretches of amino acids (⁵⁹⁶WGCSGKLICTTAVPW⁶¹⁰). The first region of flexibility (⁵⁹⁷GCSG⁶⁰⁰) is able to adopt a random coil and β -strand, as observed in the Fab 3D6-bound and F240-bound structures, respectively. The rotational freedom of the ⁵⁹⁷GCSG⁶⁰⁰ residues allows this region to be malleable to form a β -sheet with CDR-H3 when bound to Fab F240. The second region of flexibility (⁶⁰⁴CTTA⁶⁰⁷) follows a region (⁶⁰¹KLI⁶⁰³) that strongly favors the formation of an α -helix. The ⁶⁰¹KLI⁶⁰³ region is only able to form a short helix; the presence of threonine residues in ⁶⁰⁴CTTA⁶⁰⁷ tend to disfavor helix formation due to β -branching of its side chain restricting the range of rotation. We hypothesize the ⁶⁰⁴CTTA⁶⁰⁷ region destabilizes the PID α -helix to allow the PID to adopt multiple conformations. We have now added an expanded discussion on this topic (line 263-290).

5) General question related to line 107: Is this disorder in CDR-H3 involved in any way in broad selectivity? If this loops adopts several conformations, does it present disparate surfaces for interaction with a variety of epitopes? By corollary, would a structured CDR loop be more selective? It would be good if the authors could comment on this briefly.

Response: This is a very important point that the reviewer brings up. From other studies, in general, biochemical, and structural studies have shown that germline antibodies have flexible binding sites. The conformational flexibility is able to provide alternative ways of presenting the binding site to accommodate structurally unrelated ligands. Antibody maturation often reduces the flexibility and stabilizes the antibody binding site in a conformation pre-organized for interaction with the antigen, thus in turn reducing potential cross-reactivity that resulted from conformational diversity. We have added a discussion on this topic (lines 162-182).

Reviewer #2:

Cook and colleagues report the structural characterization of non-neutralizing antibodies targeting the PID on gp41. Their work contributes to the understanding of the PID misdirecting the humoral immune response at early and later stages of HIV-1 infection. Their work shows that the PID region is conformationally flexible by revealing different conformations of PID by X-ray crystallography and MD simulation alone or in complex with the close to germ-line like Ab 3D6 and by F240 that carries a higher somatic mutation rate.

The paper is well written, technically sound and provides further insight into the conformational landscape of PID. The following points need to be addressed to improve the manuscript:

1) Is there anything known about the infection state at the time of Ab isolation? Was 3D6 isolated at an early stage during infection and F240 at a later stage, explaining the difference in somatic mutations.

Response: This is an excellent question. Sadly, there is very little reported in the literature regarding the isolation of 3D6 save for it being generated by a donor with known anti-HIV antibodies in the late 1980's (Grunow et al. *J Immunol Meth* (1988) 106 257-265). The isolation of F240 was achieved through spleen-derived monocytes provided by a donor who underwent splenectomy for lymphoma staging (Cavacini, et al. *AIDS Res Hum Retrovirol.* (1998) 14, 1271-1280). We contacted the Prof. Lisa Cavacini (UMass Chan School of Medicine) and unfortunately the spleens were a part of a protocol where they received

surgically discarded tissue. There are no medical records with respect to the patient, thus, the relative stage of their infections at time of donation was unknown.

2) Line 120: please add citations to the statement:wild-type post-fusion HIV-1 gp41 via a 2.6Å hydrogen bond between S598 and S604? Furthermore, the residues should be cysteines?

Response: We appreciate identification of this typographical error – the residues should be cysteines. We have further clarified this sentence and also provided appropriate references.

“The PID region is hypothesized to form a disulfide bond between C598 and C604¹⁶. In the case of the F240-bound PID peptide, the cysteine residues were mutated to serines; a 2.6-Å hydrogen bond between C598S and C604S recapitulates the disulfide bond that would naturally exist in the PID region.”

3) Please highlight the disulfide bond (the ser positions) in the two Fab peptide structures.

Response: We have made changes to Figure 1, 2, 3 and 5 to highlight the serine positions. In Figure 1, we have drawn a disulfide linkage between the two cysteines in the Logoplot. In Figure 2, 3 and 4, C598S and C604S positions are labeled in the 2D schematic and structural figures. In addition, we have provided a description in the figure captions for clarity.

4) Throughout the manuscript, the post fusion conformation of gp41 is described as “broken gp41 spikes” or with similar terms. This should be explained and changed in the text. Consensus is that shedding of gp120 from native trimers induces the post fusion conformation of gp41 that produces the hairpin structure with both TM and Fp inserted in the same membrane, exposing PID at the tip of the rod. Appropriate citations of the gp41 post fusion conformation should be added.

Response: We thank the reviewer for this comment and agree that our use of “broken gp41 spike” can be better clarified. As such, we removed the term “broken gp41 spikes” from the text. Instead, we use the term gp41 stumps and have added an explanation on how these are formed on the surface of the virion.

“Due to the non-covalent and unstable nature of gp160, the gp120 attachment subunit is shed from the native trimers inducing a post-fusion conformation of gp41 on the surface of the mature virus^{2,3}. It has been observed that only 7-14 functional trimeric gp160 spikes exist on the virion surface^{4,5} and the majority of gp160 have lost gp120 to form fusion-incompetent gp41 stumps⁶. These post-fusion gp41 stumps produce a hairpin structure with likely both its fusion peptide and transmembrane anchor inserted into the viral membrane, thus exposing the chain reversal region.”

5) Were the MD simulations performed constraining the PID disulfide bond? If not, this could certainly contribute to the observed flexibility. Note that the post fusion conformation was suggested to exchange disulfides between protomers in the post fusion conformation (Weissenhorn et al. EMBO J 1997) in agreement with the role of disulfide exchanges during entry (Stantchev et al. Retrovirology. 2012).

Response: We thank Reviewer #2 for this comment. For the MD simulations, we did not constrain the PID disulfide bond for several reasons. First, for the smaller PID peptides, we are unable to introduce disulfide bond constraints as they were short peptide lengths, and the disulfides are near each terminus. We were concerned that this could artificially introduce bias since they would constrain the end-end distance in a small peptide which may not replicate the nature in the full protein structure. In the full HIV-1 gp41 ectodomain model, we did not constrain the disulfide bond in order to allow a comparison with the MD simulations of the PID peptides. Also as mentioned by the reviewer, the disulfide bond in gp41 appears to be labile and heterogeneous with both formation of intra- and inter-protomer disulfides and

disulfide exchange during entry (Weissenhorn W et al. *EMBO J* (1996) 15, 1507-14). Given that there are no post-fusion gp41 structures available, the positions of the cysteines in the PID loop are poorly defined.

We do appreciate the comments and position of Reviewer #2. We analyzed our MD simulations and found that the sulfur-to-sulfur distance (R-SH SH-R') was stable during the simulation (over a representative 100 ns snapshot). The sulfur-to-sulfur distance variability during the simulation was 1/10th the overall distance of the end-to-end Gly-Trp positions. Therefore, the S-S distance was highly stable without introducing further constraints. We believe with this level of stability in this important interaction, we can translate results between the full-length and smaller peptides. We have now explicitly explained this in the Results (lines 243-252) and Methods (line 547) sections.

Reviewer #3:

This is a well written manuscript with solid data that details investigations of the most immunogenic site on HIV envelope. As this immunologic site is so immunogenic, the relative paucity of knowledge of its antigenic character is surprising, so this paper adds significant knowledge to the field.

The differential presentation of this epitope however is not completely novel. Since its description, this region is known to represent more linear and conformational epitopes (Ref 6- Xu et al, *J Virol*. 1991 Sep;65(9):4832-8. doi: 10.1128/JVI.65.9.4832-4838.1991.), was explored in Ref 12 Gohain, and recently shown so in monoclonal antibody discovery (Hicar et al, *PLoS One*. 2016 Jul 13;11(7):e0158861. doi: 10.1371/journal.pone.0158861. eCollection 2016.).

Specific comments:

1) Lines 53-57 is both a bit too definitive and can use more explanation for the general audience. The PID is not completely inaccessible, as F240 in particular shows binding to SOSIP pre-fusion complexes (Kong et al, *Nat Commun*. 2016 Jun 28;7:12040. doi: 10.1038/ncomms12040.).

Response: We thank the reviewer for this comment and agree. We have softened the text and now state that the PID region is not fully accessible in the gp160 pre-fusion conformation. In addition, we have cited the *Nature Communications* reference listed by the reviewer.

2) Line 56 I think we need another sentence to explain that a significant portion of envelope on the virion surface is stubs or 'broken' env. Further explanation of "broken gp41 spikes" would clarify this as well.

Response: We agree with Reviewer #3. A similar comment was also made by Reviewer #2 and we have now added additional text to further explain "broken gp41 spikes".

"Due to the non-covalent and unstable nature of gp160, the gp120 attachment subunit is shed from the native trimers inducing a post-fusion conformation of gp41 on the surface of the mature virus^{2,3}. It has been observed that only 7-14 functional trimeric gp160 spikes exist on the virion surface^{4,5} and the majority of gp160 have lost gp120 to form fusion-incompetent gp41 stumps⁶. These post-fusion gp41 stumps produce a hairpin structure with likely both its fusion peptide and transmembrane anchor inserted into the viral membrane, thus exposing the chain reversal region."

3) RMSD and RMSF may need to be explained and distinguished better up front. Line 231 used RMSD but figure 4 RMSF.

Response: Thank you for the opportunity to clarify. RMSF stands for root mean square fluctuation. This is a numerical measurement similar to RMSD, but instead of indicating positional differences between structures, RMSF is a calculation of individual residue flexibility, or how much a particular residue

moves (fluctuates) during a simulation. RMSF per residue is typically plotted vs. residue number and can indicate structurally which amino acids in a protein contribute the most to a molecular motion. RMSD is the root mean square deviation for atomic positions and is a measure of the average distance between atoms of superimposed proteins. Thus, for comparisons of structural superimpositions, we use RMSD (Figure 4). For comparison of residue fluctuations during a molecular dynamic simulation, RMSF is commonly used. We have now added additional text to clarify this:

“The root mean-square fluctuation (RMSF), a measure of individual residue flexibility, or how much a particular residue moves during MD simulations, were 5-fold greater in the post-fusion gp41 conformation than in the pre-fusion.”